# Evaluating Real-World Robot Manipulation Policies in Simulation

**Xuanlin Li**[*1]    **Kyle Hsu**[*2]    **Jiayuan Gu**[*1]    **Karl Pertsch**[2 3 †]    **Oier Mees**[3 †]
**Homer Rich Walke**[3]    **Chuyuan Fu**[4]    **Ishikaa Lunawat**[2]    **Isabel Sieh**[2]    **Sean Kirmani**[4]
**Sergey Levine**[3]    **Jiajun Wu**[2]    **Chelsea Finn**[2]    **Hao Su**[‡1]    **Quan Vuong**[‡4]    **Ted Xiao**[‡4]

[1]UC San Diego, [2]Stanford University, [3]UC Berkeley, [4]Google Deepmind
[*]Equal contribution [†]Core contributors [‡]Co-advising

**Abstract:** The field of robotics has made significant advances towards generalist robot manipulation policies. However, real-world evaluation of such policies is not scalable and faces reproducibility challenges, issues that are likely to worsen as policies broaden the spectrum of tasks they can perform. In this work, we demonstrate that simulation-based evaluation can be a scalable, reproducible, and reliable proxy for real-world evaluation. We identify control and visual disparities between real and simulated environments as key challenges for reliable simulated evaluation and propose approaches to mitigating these gaps without the need to painstakingly craft full-fidelity digital twins. We then employ these techniques to create SIMPLER, a collection of simulated environments for policy evaluation on common real robot manipulation setups. Through over 1,500 paired sim-and-real evaluations of manipulation policies across two embodiments and eight task families, we demonstrate a strong correlation between policy performance in SIM-PLER environments and that in the real world. Beyond aggregated trends, we find that SIMPLER evaluations effectively reflect the real-world behaviors of individual policies, such as sensitivity to various distribution shifts. We are committed to open-sourcing all SIMPLER environments along with our workflow for creating new environments to facilitate research on general-purpose manipulation policies and simulated evaluation frameworks. Website: https://simpler-env.github.io

**Keywords:** Policy Evaluation, Real-to-Sim, Robot Manipulation

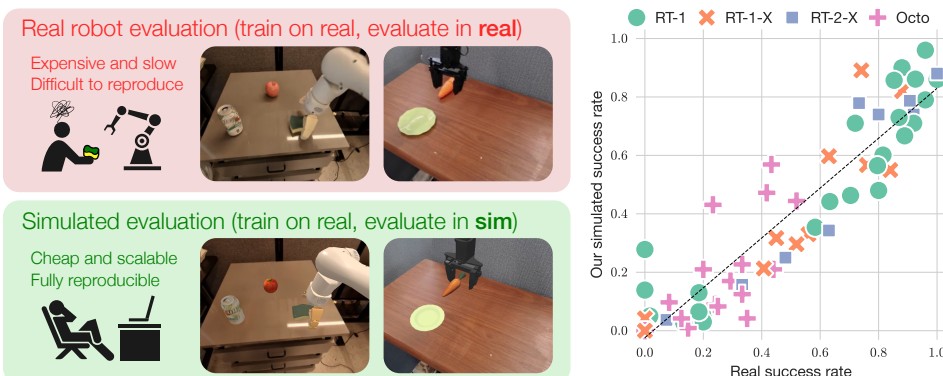

Figure 1: Characterizing generalist robot manipulation policies typically involves evaluating them on many tasks & scenarios, a laborious undertaking in the real world (top left). In this work, we design an evaluation procedure where policies trained on *real* data are evaluated in purpose-built *simulated* environments (bottom left). Our approach yields a strong correlation between real-world and simulated performance (right) for various open-source robot policies [1, 2, 3] across two commonly used robot embodiments (Google Robot and WidowX) and over 1500 evaluation episodes. These results highlight the potential of simulation-based approaches for evaluating generalist real-world robot manipulation policies in a scalable, reproducible, and reliable way.

## 1    Introduction

Remarkable progress has been made in recent years toward building generalist real-world robot manipulation policies [1, 2, 3, 4, 5], i.e., policies that can perform a wide range of tasks across many environments and even robot embodiments. However, evaluating these policies in a scalable

8th Conference on Robot Learning (CoRL 2024), Munich, Germany.

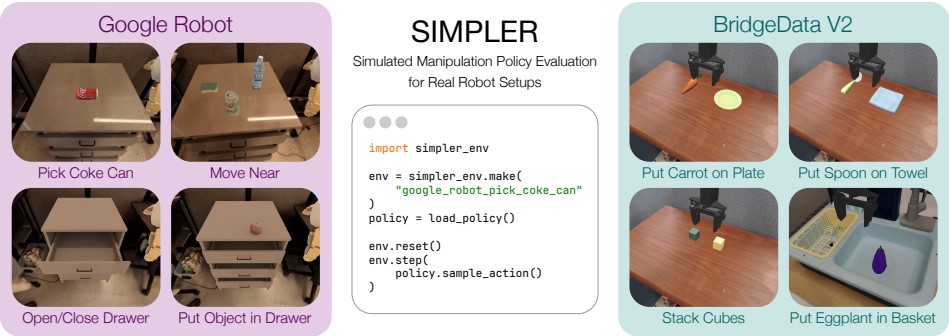

Figure 2: We introduce SIMPLER, a suite of simulated evaluation environments for common real robot manipulation setups, namely the Google Robot evaluations from the Robotics Transformer series of works [1, 2, 17], and environments from the BridgeData V2 dataset [5] (see Appendix for detailed environment descriptions). We make using SIMPLER easy via exposing a standard Gym interface. Additionally, we commit to open-source policy inference code for real-to-sim evaluation of common generalist robot policies (RT-1 [1], RT-1-X [2], and Octo [3]), and provide detailed guides for evaluating new policies and creating new evaluation environments.

and reproducible way remains challenging, as real-world evaluation is expensive and inefficient. Compared to the evaluation burden of works that study robot performance in narrower settings, the scope of evaluations required for faithful performance estimates of *generalist* policies increases with the breadth of their abilities. This underlines a growing challenge in robot manipulation research: as we scale the capabilities of robot policies, how do we correspondingly scale our ability to accurately, reproducibly, and comprehensively evaluate them?

In this work, we propose *simulated evaluation* as a possible answer, in which manipulation policies trained on real data are evaluated in purpose-built simulated environments (Fig. 1). Such real-to-sim evaluation can serve as a scalable, reproducible, and informative tool to complement gold-standard real-world evaluations. Indeed, evaluation in simulation is common practice for testing autonomous driving policies across a wide range of scenarios before real-world deployment [6, 7, 8]. However, performing simulated evaluations for robotic *manipulation* poses additional challenges due to the diverse interactions between the agent and the environment. At the same time, research on sim-to-real policy learning [9, 10] has demonstrated that considerable transfer between simulation and the real world is possible even for manipulation policies. While sim-to-real approaches typically train in simulation and evaluate in the real world, we are interested in the opposite question: how can we build systems for evaluating manipulation policies trained on *real* data in *simulated* environments?

One option to build such simulated environments is to fully replicate an existing real-world environment by creating a simulated "digital twin", an approach popular in navigation [11, 12] and autonomous driving [13]. However, for robot manipulation, reconstructing dynamic and interactive objects [14], along with realistic materials and lighting [15] in simulation remains an open research question. Furthermore, building full-fidelity digital twins demands extensive time and resources, typically requiring digital artists to manually craft object geometries and materials [11, 16]. Capturing precise physical properties of objects for manipulation simulation, such as the center of mass, inertia, and static and dynamic friction, further complicates scalability.

A key idea in this work is that we do not need to build *exact* replicas of real-world environments. Instead, we aim for simulated environments that are merely *realistic enough*, such that the performance of policies evaluated in simulation correlates well with their real-world performance. This allows us to design environment creation pipelines that are more scalable than creating exact digital twins. Through extensive experiments, we examine the challenges of building effective simulated evaluation pipelines, specifically control and visual disparities between real and simulated environments. We then propose and evaluate approaches for mitigating these differences based on offline system identification, "green-screening" simulation observations using real-world backgrounds, and object texture baking from real-world images.

Using these techniques, we create SIMPLER, a suite of simulated manipulation policy evaluation environments for commonly used real robot setups, namely the RT-1 [1] and BridgeData V2 [5]

evaluation setups (Fig. 2). For both setups, we perform extensive *paired* sim-and-real evaluations for multiple open-source manipulation policies such as RT-1-X [2] and Octo [3], and we demonstrate a strong correlation between policy performance as assessed by SIMPLER and the corresponding real environments (Fig. 1, right). In addition, we find that simulated policy evaluations in SIM-PLER environments accurately reflect policy behavior modes in the real world, such as sensitivity to various distribution shifts. As such, SIMPLER is a first step toward using simulated evaluation as a tool for reliable, scalable, and reproducible manipulation policy evaluation. We are committed to open-sourcing our workflow for constructing SIMPLER environments to facilitate research on general-purpose manipulation policies and simulated evaluation frameworks.

## 2   Related Work

Reproducible evaluation of real robot policies is a long-standing challenge. Initiatives like YCB [18] and NIST [19] were introduced to standardize evaluation objects, yet standardizing other variables such as lighting, camera setups, and workspaces proves difficult. Other benchmarks like the Amazon Picking Challenge [20], DARPA Robotics Challenges [21], TOTO [22], and RB2 [23] require ongoing maintenance of real-world evaluation setups, representing a significant long-term investment. As real-world robotic datasets [1, 2, 5] and generalist policies [24, 17, 3, 25] proliferate, the demand for *reliable, scalable, and reproducible* methods of evaluating these policies grows. The need is particularly acute given the difficulty faced by the research community in conducting evaluations without standardized hardware.

Simulation-based algorithmic research offers an alternative to real-world evaluation. A wide range of simulation benchmarks [26, 27, 28, 29, 30, 31, 32, 33, 34, 35, 36, 37, 38] have been established to facilitate scalable and reproducible evaluation. However, most previous work considers both training and evaluation in simulation, and the resulting policies might exhibit distinct behaviors when deployed on real robot hardware. In contrast, we aim to both measure and enhance simulated evaluations' ability to reflect a policy's real-world performance and behaviors.

Can simulated evaluations reliably predict real-world policy performance and behavior modes? While several studies [12, 11, 39] have explored this question in navigation tasks, we focus on simulation evaluation for real-world *manipulation* policies. Manipulation presents unique challenges due to the tight interaction loop between policy and environment, dynamic rather than static scenes, along with the intricate, precise, and complex action sequences where even slight variations can significantly impact task outcomes.

For robot manipulation, the sim-to-real setting has been extensively investigated, where one aims to train policies in simulation and deploy them in the real world. To tackle the gaps between simulation and reality, prior work has adopted domain randomization [40, 41, 42, 43, 44, 10, 45, 46, 47] and domain adaptation [48, 49, 50, 51, 52] approaches. For instance, Generative Adversarial Networks (GANs) [48, 49, 50, 51] are trained to modify images generated in simulations so they resemble the style of real-world images. Alternatively, Du et al. [52] aim to align the feature space of observations between simulated and real-world environments, creating a more consistent visual experience across these domains. In contrast to these methods of sim-to-real learning, we focus on the opposite question: *building simulation systems that effectively and faithfully evaluate real-world robot manipulation policies*. To this end, we introduce approaches to addressing both real-to-sim visual and control gaps to enhance real-&-sim evaluation correlations.

## 3   Using Physics Simulators for Evaluation of Robot Manipulation Policies

**Problem formulation.** We study the problem of using physics simulators to evaluate performance and examine the behavior modes of real robot manipulation policies. We emphasize that our goal is *not* to completely replace real-world evaluations or to perfectly replicate real-world policy behaviors in simulations, as there are always gaps between sim and real. Instead, we aim to achieve strong *correlation* between *relative* policy performances in the real world and in simulation. This gives practitioners a readily available policy improvement signal to guide their research. Formally, for two policies $\pi_a$ and $\pi_b$ with real-world performance measures $R_a$ and $R_b$ - such as their success rates across tasks - we aim to construct a simulator $\mathcal{S}$ (with performance measures $R_{\mathcal{S},a}$ and $R_{\mathcal{S},b}$) that achieves strong *correlation* between relative performances in simulation and in the real world.

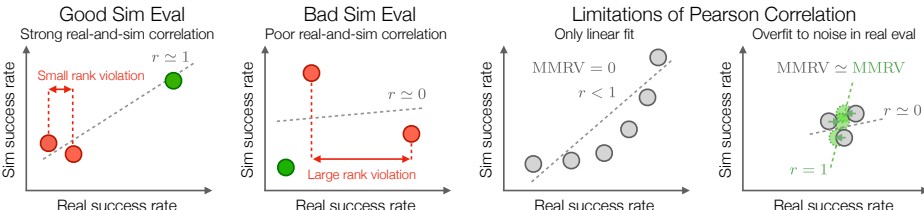

Figure 3: Illustration of mean maximum rank violation (MMRV, range $[0, 1]$, lower is better) and Pearson correlation coefficient (Pearson $r$, range $[-1, 1]$, higher is better) for assessing the policy performance correlation in real-world and simulation, as well as the overall quality of simulated evaluation pipelines. Each circle represents a policy. For the two leftmost pipelines, both metrics yield valuable insights, identifying one as poor and the other as good. The two rightmost examples highlight the limitations of Pearson $r$: it can penalize simulation pipelines that fail to linearly recover the real results despite recovering the correct ranking, and it is overly sensitive to minor noise in evaluations when different policies perform similarly in the real world.

**Metrics for real-to-sim evaluation pipelines.** A standard approach for measuring the correlation between two variables is the **Pearson correlation coefficient (Pearson $r$)** [53]. However, Pearson correlation has two important limitations for judging simulated evaluation pipelines: **(1)** It only assesses the *linear* fit between real and simulated performance, which is not necessarily required so long as the simulated pipeline reflects real-world performance improvements between different policies (Fig. 3, middle right); **(2)** It does not reflect the *range* of values it is computed over. Thus, for policies performing similarly in the real world, $r$ may change drastically based on the inherent noise in real-world evaluations (Fig. 3, far right grey vs. green).

To address the first drawback of the Pearson correlation, we can additionally report a *ranking* metric that measures whether the simulated evaluation ranks the policies correctly based on their real-world performance, independent of whether the relationship is linear. However, conventional ranking metrics such as Spearman's rank correlation [54] still suffer from the second shortcoming: they operate purely on the rankings and disregard the underlying margins between real values. When choosing a suitable ranking metric, the key point is that we need to take the *magnitude* of the rank violation into account, measured as the difference in real-world performance between the mis-ranked policies. This provides a clear signal whether rank violations are caused by small real-world performance differences that are often the result of inherent noise in real robot evaluations, as the far-left example of Fig. 3, or constitute clear failures of the evaluation pipeline, as the middle-left example of Fig. 3.

Thus, we propose the **mean maximum rank violation (MMRV)** metric to better assess the consistency of the real-and-sim policy ranking. Given $N$ policies $\pi_{1...N}$ and their respective performance measures (e.g., success rates) $R_{1...N}$, $R_{\mathcal{S},1...N}$ from real and simulated evaluations, we have:

$$\text{MMRV}(R, R_{\mathcal{S}}) = \frac{1}{N} \sum_{i=1}^{N} \max_{1 \leq j \leq N} |R_i - R_j| \, \mathbf{1}[(R_{\mathcal{S},i} < R_{\mathcal{S},j}) \neq (R_i < R_j)] \quad (1)$$

The key underlying quantity is the *rank violation* between two policies $\pi_i$ and $\pi_j$, which weighs the significance of the simulator's incorrect ranking by the corresponding real-world performance margin. MMRV then averages each policy's worst-case rank violation. We will report both MMRV and Pearson $r$ as they provide complementary perspectives on the simulated evaluation's effectiveness.

## 4 SIMPLER: Simulated Manipulation Policy Evaluation for Real Robots

This section introduces our approach to designing a simulation evaluation pipeline for real robot manipulation policies. We take inspiration from the rich literature on sim-to-real policy learning [45, 40, 41, 10] and focus on control and visual gaps between simulation and the real world.

### 4.1 Mitigating the Real-to-Sim Control Gap via Offline System Identification

First, we need to ensure that the policy actions executed in simulation yield comparable effects on the system state as those observed when executed on the real robot. Concretely, let $\{(\mathbf{x}_i, R_i) : \mathbf{x}_i \in \mathbb{R}^3, R_i \in \mathbb{SO}(3)\}_{i=1}^T$ be a 6D end-effector pose trajectory recorded when rolling out an action trajectory $\{\mathbf{a}_i\}_{i=1}^T$ on a real robot. Let $\{(\mathbf{x}'_i, R'_i) : \mathbf{x}'_0 = \mathbf{x}_0, R'_0 = R_0\}_{i=1}^T = \text{Sim}(\mathbf{p}, \mathbf{d}, \{\mathbf{a}_i\}_{i=1}^T, \mathbf{x}_0, R_0)$ be the corresponding trajectory when unrolling the same sequence of actions in the simulation in an *open-loop* manner using stiffness and damping parameters (i.e., PD parameters) $(\mathbf{p}, \mathbf{d})$. Then, we define our system identification loss from translation and rotation errors as $\mathcal{L}_{\text{sysid}}(\mathbf{p}, \mathbf{d}) = \frac{1}{T} \sum_{i=1}^T \left( ||\mathbf{x}_i - \mathbf{x}'_i||_2 + \arcsin\left(\frac{1}{2\sqrt{2}}||R_i - R'_i||_F\right) \right)$.

In practice, we use a small sample of 20 trajectories from an offline dataset $\mathcal{D}$, e.g., a real-world demonstration dataset, to retrieve action and end-effector pose trajectories and compute the system identification loss above. For all environments in this work, we use trajectories from existing open-source demonstration datasets [1, 5], and thus do not need to collect any new data. We also ensured that each trajectory comes from different task instructions to encourage trajectory diversity.

Next, we optimize the parameters of our controller: given initial PD parameters $(\mathbf{p}_0, \mathbf{d}_0)$ and search ranges $[\mathbf{p}_{\text{low},0}, \mathbf{p}_{\text{high},0}], [\mathbf{d}_{\text{low},0}, \mathbf{d}_{\text{high},0}]$, we normalize the range to $[0, 1]$ and perform simulated annealing [55] to optimize $\mathcal{L}_{\text{sysid}}$ in a gradient-free manner. We then select the PD parameters with the lowest $\mathcal{L}_{\text{sysid}}$ as $(\mathbf{p}_1, \mathbf{d}_1)$, and initialize another round of simulated annealing with a reduced parameter search range. In total, we perform 3 rounds of simulated annealing.

On the right, we qualitatively illustrate the effects of our system identification for one of our simulated environments, the Google Robot [1]. We find that naively using PD parameter values from real controllers results in inaccurate track-

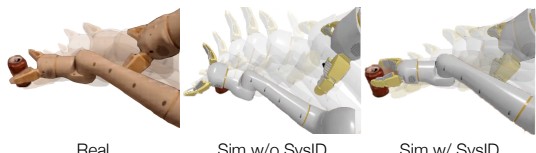

Real      Sim w/o SysID      Sim w/ SysID

ing of the real robot's end-effector movements, which culminates in a missed grasp on the Coke can. After system identification, the controller more accurately tracks the motion in simulation: the robot is able to grasp the object when replaying the demonstration's action sequence.

## 4.2   Mitigating the Real-to-Sim Visual Gap via Visual Matching and Variant Aggregation

Visual discrepancies between real and simulated environments can cause distribution shifts that adversely impact policy behavior, making simulated evaluations less reliable [11]. While modern graphics pipelines are able to create highly realistic visuals, developing the underlying assets and determining the lighting parameters to accurately model existing environments involves significant manual labor. Our goal is to match the simulator visuals to those of the real-world environment with only a *modest* amount of manual effort. For the scene background, we

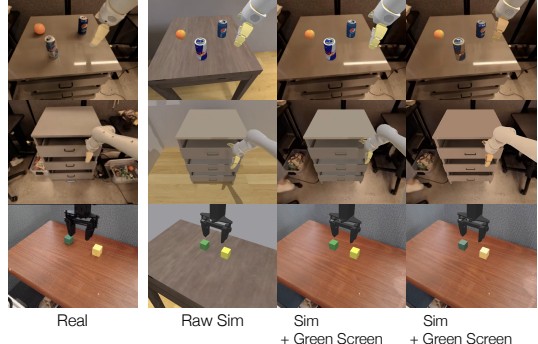

Real      Raw Sim      Sim + Green Screen      Sim + Green Screen + Texture Matching

propose a "green screening" approach in which we overlay an image of the real-world environment onto the background of the simulated scene (see the wrapped figure). Concretely, we perform the following steps: (1) we remove the robot and foreground objects from the first frame of a real-world evaluation video $I_{\text{real}}$ using online image inpainting tools (e.g., https://cleanup.pictures); (2) we create a binary mask $M$ isolating the foreground objects (robot arm and interactable objects, such as tabletop objects and articulated objects) in the simulation rendering $I_{\text{sim}}$ by querying ground truth segmentation masks in simulation; and (3) we combine the real-world background with the simulation foreground: $I' = M \odot I_{\text{sim}} + (1 - M) \odot I_{\text{real}}$, producing the green-screened image.

In practice, we find that many tested policies are also sensitive to changes in foreground object and robot textures, yet available simulation assets often exhibit appearance differences from real-world objects. Thus, we tune simulation asset textures for many objects to more closely match their real-world counterparts. We term this approach **"Visual Matching"** (see above for an illustration). Concretely, we project the real texture onto the simulation object by (1) segmenting the object in a real-world image [56]; (2) aligning the simulated object pose to the real image; and (3) "unprojecting" the texture onto the object mesh in simulation. We provide step-by-step instructions in Appendix B.2, and we will release a convenient command-line script for this process. Otherwise, for assets like robot visual meshes with texture maps already resembling their real-world counterparts, we can instead selectively copy and paste color values from real to simulated texture maps. Finally, as robot arms may change colors during movement, we found it helpful to obtain multiple tuned robot arm colors that match the real-world textures from different phases of a manipulation task. We then average their evaluation results to mitigate this confounding factor.

As an alternative to Visual Matching, we explore a mitigation strategy for real-to-sim visual gaps inspired by domain randomization: instead of minimizing the gap, we heavily randomize visual aspects of the scene to create environment variants. We then obtain an estimate of a policy's performance by aggregating evaluation results across multiple such variants, which we term as **"Variant Aggregation"** (see Appendix B.3 for more details and visualizations). Note that the Variant Aggregation setup does not seek to match the visual appearances between sim and real. As a result, it can utilize any scenes from public datasets like ReplicaCAD [16] and any objects (including the tabletop objects and the cabinets) from public datasets like Objaverse [57] for evaluation, whose appearances are out-of-distribution for the policies evaluated in this paper.

**SIMPLER environments and other details**. We instantiate SIMPLER on two commonly used real robot evaluation setups: the Google Robot from the RT series of work [1, 17, 2] and the WidowX BridgeData V2 setup [5]. For each setup, we provide simulations for multiple tasks spanning a range of skills, interacted objects, object positions and orientations, backgrounds, and lighting conditions (see Fig. 2). The tasks are chosen to be representative of those in the corresponding training datasets, while also involving largely rigid body objects whose dynamics can be reasonably well-approximated by modern physics simulators. We also instantiate SIMPLER on top of the SAPIEN physics simulator [58], but later show that our contributions are independent of the simulator used and can be reproduced in Isaac Sim [59]. Creating most new environments takes an experienced user approximately one hour. See Appendix A and Appendix B for more details.

## 5 Experimental Results

In this section, we empirically test the performance correlation between real-world robot evaluations and simulated evaluations in SIMPLER environments for a representative set of open-source generalist robot manipulation policies. Concretely, our experiments are designed to answer the following questions: **(1)** Do relative performances of different manipulation policies in SIMPLER **strongly correlate** with those observed in real evaluation (Section 5.2)? **(2)** For a given policy, can SIMPLER evaluations accurately reproduce real-world policy behavior modes, e.g. sensitivity to various **visual distribution shifts** (Section 5.3)? **(3)** How sensitive is the fidelity of SIMPLER evaluations to (i) control and visual gaps, (ii) physical property gaps, and (iii) the choice of simulator (Section 5.4)?

### 5.1 Experimental Setup

To quantitatively evaluate correlations between real and simulation policy performance, we perform paired sim-and-real experiments, i.e., with the same task instructions, object & robot poses, evaluation trials, and success criterions between the simulation and the real world (note that the paired experiments are not restricted to the tasks and setups in policies' training data). We use popular open-source generalist robot policies: RT-1-X [2] and Octo [3] (Octo-Base and Octo-Small). For evaluations in the Google Robot environments, we additionally use a number of RT-1 [1] checkpoints at various stages of training: RT-1 trained to convergence (RT-1 (Converged)), RT-1 at 15% of training steps (RT-1 (15%)), and RT-1 at the beginning of training (RT-1 (Begin)). We also report results on RT-2-X [17]. More details in Appendix A.

### 5.2 SIMPLER Environments Show Strong Performance Correlations with Real Evaluations

We summarize the results of our main paired real-world and simulation evaluations in Fig. 4 (detailed breakdown in Appendix Table 2). We observe a strong correlation between the relative performances in simulation and in the real world across most policy checkpoints and tasks we evaluate, as reflected in low values for the MMRV metric introduced in Section 3 and high values for Pearson $r$.

In Table 1, we compare SIMPLER with using the action MSE in validation episodes for policy ranking, which is common for model selection within supervised learning settings (e.g., imitation learning). However, we find that validation MSE is *not* a good proxy for a policy's real-world performance, leading to high MMRV and low Pearson $r$. On the other hand, SIMPLER evaluations more accurately reflect relative policy performances in the real world, obtaining significantly lower MMRV and higher Pearson $r$. Additionally, we find that an alternative implementation of SIMPLER using Variant Aggregation (Section 4.2) performs worse: since it does not seek to match real-world visuals (Fig. 5), policies more sensitive to visual distribution shifts exhibit larger real-and-sim per-

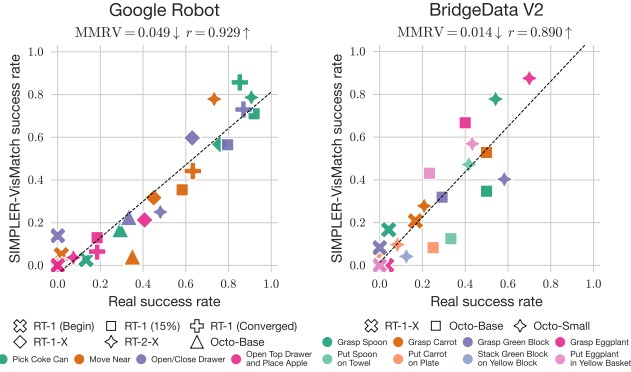

Figure 4: Real vs. SIMPLER success rates on Google Robot & Bridge V2 tasks. SIMPLER evaluations ("Visual Matching" setup) show a strong correlation with real policy performance.

| Evaluation Protocol | Pick Can | Move Near | Drawer | Avg. |
|---|---|---|---|---|
| | MMRV ↓ | | | |
| Validation MSE | 0.412 | 0.408 | 0.306 | 0.375 |
| SIMPLER-VarAgg | 0.084 | **0.111** | 0.235 | 0.143 |
| SIMPLER-VisMatch | **0.031** | **0.111** | **0.027** | **0.056** |
| | Pearson $r$ ↑ | | | |
| Validation MSE | 0.464 | 0.230 | 0.231 | 0.308 |
| SIMPLER-VarAgg | 0.960 | **0.887** | 0.486 | 0.778 |
| SIMPLER-VisMatch | **0.976** | 0.855 | **0.942** | **0.924** |

Table 1: Comparison of manipulation policy evaluation protocols for ranking 6 common open-source policy checkpoints (3 RT-1 checkpoints, RT-1-X, RT-2-X, Octo-Base) on Google Robot tasks. See Appendix Table 2 for a detailed breakdown of results per policy.

formance discrepancies. In summary, **SIMPLER leads to a strong correlation with real-world policy performance**, and we recommend Visual Matching as the default evaluation approach.

### 5.3 SIMPLER Evaluations Effectively Model Policy Robustness to Distribution Shifts

Previously, we showed that SIMPLER evaluations strongly correlate with real-world performances based on trial averages. Beyond comparing average policy performances, it would be beneficial to let practitioners gauge more fine-grained aspects of a policy's behavior, such as its robustness to distribution shifts like lighting, background, and texture changes. We ask: do SIMPLER evaluations accurately reflect a policy's real-world behavior under such distribution shifts, and can they thus be used for more fine-grained policy evaluation beyond average performance?

To test this, we use SIMPLER environments to perform controlled experiments along five distribution shift axes inspired by Xie et al. [60]: background, lighting, distractors, table texture, and robot camera pose. We adopt the base environment setup and the two variations per axis from our Variant Aggregation evaluation (see Appendix B.3), adding two more variations for the

new camera pose axis. We evaluate two RT-1 checkpoints trained with and without data augmentation, which exhibit different robustness behaviors to distribution shifts. For simulated results and real-world results, we report the difference in the success rate with and without each distribution shift. The wrapped figure shows the results (more details in App. Table 4). We find that SIMPLER evaluations effectively reflect the policies' robustness to various distribution shifts in the real world. Notably, in both real and sim, changing robot

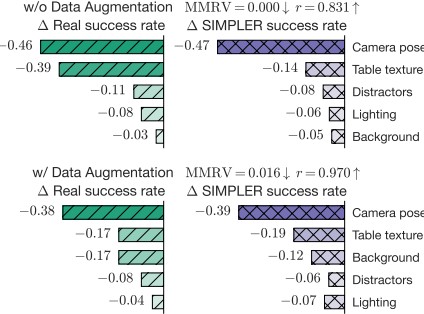

camera poses and table textures has a significant impact on policy performance, while the impact of lighting and distractor changes are relatively minor.

Furthermore, we find that an even more fine-grained analysis is possible through simulated evaluation. For example, when varying real-world table textures, both policies are more robust to unseen *solid* table colors than unseen *patterned* table textures (4% vs. 25% avg. performance decrease). This behavior is well reflected in SIMPLER: policy performance in simulation decreases by 2% on average when new colors of the solid table are introduced and by 24% for new patterned textures.

**Testing novel distribution shifts.** Based on these results, we put our simulated evaluations to the test and ask: can SIMPLER evaluations be used to *predict* the robustness of policies to new distribution shifts in the real world? Throughout our simulated evaluations, we observe that Octo-Base is particularly sensitive to changes in the simulated robot arm textures. Specifically, under our "Visual Matching" evaluation setup, its success rate is 0% on the "Pick Coke Can" task using the un-

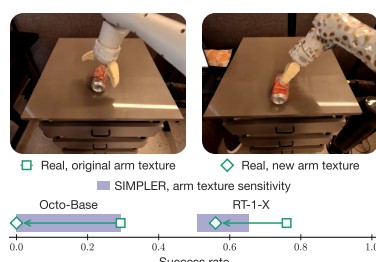

tuned robot arm, but 29.3% using one of our tuned robot arms. On the other hand, RT-1-X, also trained on the same Open-X-Embodiment dataset [2], exhibits higher robustness to different simulated robot arm textures. To test whether this trend in simulation holds in real-world evaluations, we design a novel real-world distribution shift evaluation, where we change the real robot arm texture by wrapping it using multiple gift wrapping papers. We report results in the wrapfigure and Appendix Table 6. The real-world evaluations support the simulated results: Octo-Base is more sensitive to changes in arm texture than RT-1-X. This indicates that simulated evaluations in SIMPLER environments can be predictive of real-world policy behaviors under novel distribution shifts.

## 5.4 Ablation Studies

**Effect of system identification.** To test the impact of system identification (Section 4.1), we repeat the Google Robot's Visual Matching evaluations from Section 5.2, but perturb the stiffness and damping parameters of robot joints from the results of system identification.

| Control Params | Control Loss ↓ | MMRV ↓ |
|---|---|---|
| Setting 1 | 0.267 | 0.070 |
| Setting 2 | 0.432 | 0.100 |
| SIMPLER SysID | **0.131** | **0.031** |

On the right, we show that the noisy system identification parameters lead to worse MMRV, i.e., worse correlation between simulated and real-world evaluations. This underlines the importance of accurate system identification for simulated evaluation.

**Effect of visual matching.** We ablate the impact of our approaches in Section 4.2 for matching sim and real visual appearances. We use the RT-1 (Converged), RT-1 (Begin), and RT-1-X checkpoints on the Google Robot's open/close drawer tasks, and we compare the effectiveness of different combinations of background "green-screening", object texture, and robot texture settings. Results are reported in the wrapfigure

| Green Screen | Drawer Matching | Robot Matching | MMRV ↓ | Real-Sim Success Gap ↓ |
|---|---|---|---|---|
| ✗ | ✗ | ✗ | 0.087 | 0.272 |
| ✗ | ✓ | ✗ | 0.087 | 0.266 |
| ✗ | ✗ | ✓ | 0.087 | 0.272 |
| ✗ | ✓ | ✓ | 0.087 | 0.328 |
| ✓ | ✗ | ✗ | 0.087 | 0.198 |
| ✓ | ✓ | ✗ | 0.142 | 0.253 |
| ✓ | ✓ | ✓ | **0.050** | **0.136** |

(more details in App. Table 7). We observe the lowest MMRV and real-to-sim performance gap when combining background "green-screening" with object texture tuning for *both* drawer and robot assets. Interestingly, only tuning the drawer texture or the robot texture (but not both) is insufficient in improving the real-and-sim correlation. We hypothesize that this causes appearance *inconsistencies* between different parts of a scene, resulting in larger real-and-sim performance disparities.

**Sensitivity to physical property gap.** When developing SIMPLER environments, we simplified the physical properties (e.g., the center of mass and friction coefficients) of objects and robots due to the complexity and time-consuming nature of precise modeling and system identification. In this section, we investigate whether our simulated evaluation is sensitive to such a real-to-sim physical property gap. We conduct 2 experiments: **(1)** For the "pick coke can" task, we vary the mass of the empty coke can (by varying its density), along with the static friction of the gripper finger; (2) For the "open/close drawer" task, we vary the joint frictions of the articulated cabinet. We report the MMRV and the Pearson correlation results in Appendix Tab. 8. We find that our simulated evaluation remains effective across a spectrum of plausible physical property parameters, evidenced by the low MMRV and the high Pearson correlation, even though altering these parameters has a moderate ($\leq 15\%$) impact on the success rates of different policies.

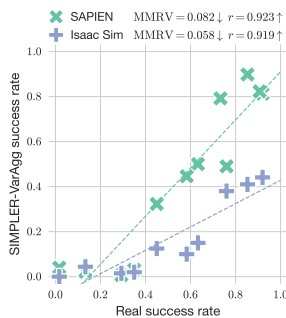

**Sensitivity to the choice of physics simulator.** To investigate whether our results are sensitive to the underlying physics simulator, we reproduce the Google Robot evaluation in Isaac Sim [59]. The results on the right (details in Table 9) show that SIMPLER's results are reproducible in Isaac Sim. In particular, we also observe a strong real-to-sim performance correlation across most checkpoints for SIMPLER-Isaac. This suggests that the choice of physics simulator is not critical for the tasks we tested, which mainly involve rigid body manipulations.

## 6 Conclusion

As generalist robot manipulation policies advance, scalable evaluation approaches become essential for the rapid development of algorithms, models, and datasets. We introduce SIMPLER, a suite of real-to-sim evaluation environments that show a strong correlation to real-world policy performance. We provide a detailed discussion of our current work's limitations in Appendix Sec. G.

**Acknowledgments**

We sincerely thank Jeffery Bingham and Paul Wohlhart from Google DeepMind for clarifying some details of the RT-1 Robot controller. We also thank Justice Carbajal, Samuel Wan, Jornell Quiambao, Deeksha Manjunath, Jaspiar Singh, Sarah Nguyen, Jodilyn Peralta, and Grecia Salazar for conducting real-world RT-1 Robot experiments. We thank Google DeepMind for the RT-2-X alpha access. Additionally, we thank Fanbo Xiang from UC San Diego for his help in matching the real and simulation visual appearances of foreground objects. Kyle Hsu was supported by a Sequoia Capital Stanford Graduate Fellowship.

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

# A  Full Environment and Evaluation Protocol Details

In this section, we provide detailed descriptions of our SIMPLER environments along with our simulation and real-world evaluation protocols.

For the RT-1 Robot, we adopt the following language-conditioned tasks:

- **"pick coke can"**. The robot is instructed to grasp the empty coke can on the table and lift it up. In the default setting, no distractors are added to the scene. We place the coke can in 3 different orientations: horizontally laying, vertically laying, and standing. For each orientation, we place the coke can at 25 grid positions within a rectangle on the tabletop, yielding 25 trials per orientation and 75 trials in total.

- **"move {obj1} near {obj2}"**. We place a triplet of objects on the tabletop in a triangle pattern. In each trial, one object serves as the source object, one serves as the target, and the other serves as the distractor (this creates 6 trials for each triplet and each triangle pattern). We randomly choose 5 triplets of objects among a total of 8 objects (blue plastic bottle, pepsi can, orange, 7up can, apple, sponge, coke can, redbull can), and adopt 2 triangle patterns (upright and inverted). This creates a total of $5 \times 2 \times 6 = 60$ trials. The 5 triplets chosen are:
  - blue plastic bottle, pepsi can, orange
  - 7up can, apple, sponge
  - coke can, redbull can, apple
  - sponge, blue plastic bottle, 7up can
  - orange, pepsi can, redbull can

- **"(open / close) (top / middle / bottom) drawer"**. The robot is positioned in front of a cabinet that contains 3 drawers and instructed to open / close a specific drawer, testing its ability to manipulate articulated objects. We place the robot at 9 grid positions within a rectangle on the floor, yielding a total of $9 \times 3 \times 2 = 54$ trials.

- **"open top drawer; place apple into top drawer"**. The robot opens the top drawer and places the apple from the cabinet top into the top drawer, testing its ability to perform longer-horizon tasks. We place the robot at 3 different positions on the floor and the apple at 9 different positions within a grid on the cabinet top, yielding a total of $3 \times 9 = 27$ trials. Initially, the policies receive the "open top drawer" instruction. We switch to the "place apple into top drawer" instruction once the robot outputs the "terminate" token or after half of the time limit has elapsed.

For the WidowX + Bridge (with WidowX-250 6DOF robot), we adopt the following tasks:

- **"put the spoon on the towel"**. We place the spoon on a vertex of a square (with edge length 15cm) on the tabletop, and we place the towel on another vertex. The spoon's initial orientation switches between horizontal and vertical, requiring the robot to perform gripper reorientation. This creates a total of $2 \times 12 = 24$ trials.

- **"put carrot on plate"**. We adopt a similar setup as "put the spoon on the towel", replacing the spoon with carrot and the towel with plate.

- **"stack the green block on the yellow block"**. We place a green block on a vertex of a square on the tabletop, and we position a yellow block on another vertex. The block dimensions are 3cm. We also adopt two differently-sized squares (edge length 10cm and 20cm). This creates a total of $2 \times 12 = 24$ trials.

- **"put eggplant into yellow basket"**. We place an eggplant on the right basin of a sink, and we place a yellow basket on the left basin. The eggplant is dropped into the sink at a random position and orientation, and we ensure that the eggplant is directly graspable (i.e., not too close to the edges of the sink basin). We perform a total of $24$ trials.

**Algorithm 1** RT-1 Robot Controller in Simulation

---

**Require:** (1) Current end-effector action $(\mathbf{x}_a, R_a)$, along with sensed arm joint positions and velocities $q_{\text{arm}}, v_{\text{arm}}$; (2) Current gripper action $g_a$, along with sensed gripper joint position and velocity $q_{\text{grip}}, v_{\text{grip}}$; (3) Simulation frequency $H_{\text{sim}}$ (501 in our implementation), action output frequency (control frequency) $H_{\text{ctrl}}$ (3 in our implementation following [1]); (4) Arm velocity, acceleration, and jerk limits $L_{\text{arm}}$ (equal to 1.5, 2.0, 50.0 respectively); (5) Gripper velocity, acceleration, and jerk limits $L_{\text{grip}}$ (equal to 1.0, 7.0, 50.0 respectively); (6) Current action timestep $T$ within an episode; (7) A planner that takes goal and initial joint positions and velocities as input (along with velocity, acceleration, and jerk constraints), and outputs a time-parametrized trajectory.

1: # Arm motion planning
2: $(\mathbf{x}, R) = \text{ForwardKinematics}(q_{\text{arm}})$
3: $(\mathbf{x}_{\text{goal}}, R_{\text{goal}}) = (\mathbf{x}_a + \mathbf{x}, R_a \cdot R_{\text{arm}})$
4: $(q_{\text{goal}}, v_{\text{goal}}) = (\text{InverseKinematics}(\mathbf{x}_{\text{goal}}, R_{\text{goal}}, q_{\text{arm}}), 0.0)$
5: ArmPlan = $\text{Planner}(q_{\text{goal}}, v_{\text{goal}}, q_{\text{arm}}, v_{\text{arm}}, L_{\text{arm}})$
6: # Gripper motion planning
7: **if** $T = 0$ **then**                                            ▷ At the beginning of episode
8:     $q_{\text{lastplan,grip}}, v_{\text{lastplan,grip}} = q_{\text{grip}}, 0.0$
9:     $q_{\text{lastgoal,grip}} = q_{\text{grip}}$
10: **end if**
11: **if** $|g_a| < 0.01$ **then**                                            ▷ Small action filtering
12:     $q_{\text{goal,grip}} = q_{\text{lastgoal,grip}}$
13: **else**
14:     $q_{\text{goal,grip}} = q_{\text{lastplan,grip}} + g_a$
15: **end if**
16: $v_{\text{goal,grip}} = 0.0$
17: GripPlan = $\text{Planner}(q_{\text{goal,grip}}, v_{\text{goal,grip}},$
                        $q_{\text{lastplan,grip}}, v_{\text{lastplan,grip}}, L_{\text{grip}})$
18: # Execute arm and gripper plans at each simulation step
19: **for each** $i = 1 \cdots \frac{H_{\text{sim}}}{H_{\text{ctrl}}}$ **do**
20:     $t = \frac{i}{H_{\text{sim}}}$
21:     $q_{\text{lastplan}}, \_ = \text{ArmPlan}(t)$
22:     $\text{SetArmJointPosTarget}(q_{\text{lastplan}})$
23:     $q_{\text{lastplan,grip}}, v_{\text{lastplan,grip}} = \text{GripPlan}(t)$
24:     $\text{SetGripperJointPosTarget}(q_{\text{lastplan,grip}})$
25:     $\text{SetGripperJointVelTarget}(v_{\text{lastplan,grip}})$
26: **end for each**
27: $q_{\text{lastgoal,grip}} = q_{\text{goal,grip}}$
28: $T = T + 1$

---

For Octo simulated evaluations, since the model involves a non-deterministic diffusion head, we average its success rates across three different random seeds to produce a lower-variance estimate of the policy's simulation performance. Additionally, for RT-1 Robot simulated evaluations, we average results over four versions of robot arm and gripper colors to account for changes in arm texture during real robot rollouts (see Section 4.2). For the WidowX environments, given the consistent black color of the arm and gripper across videos, we skip this step.

The number of evaluation trials we present above pertain to the real-world evaluation setup. For our "Variant Aggregation" simulation evaluation setup, the number of trials is multiplied by the number of simulation environment variants. For our "Visual Matching" simulation evaluation setup, the number of trials is multiplied by the number of tuned robot arm colors for the RT-1 Robot evaluation setup, along with the number of seeds for the Octo policies.

## B  More Implementation Details of Our Real-to-Sim Evaluation System

### B.1  Robot Controllers

**RT-1 Robot** Given translation, rotation, and gripper action output from a model, we adopt Algorithm 1 in simulator to execute the action commands. The simulation frequency in the algorithm refers to the number of simulation steps per second, while the control frequency refers to the num-

---

**Algorithm 2** WidowX Controller in Simulation

---

**Require:** (1) Current end-effector action $(\mathbf{x}_a, R_a)$, along with sensed arm joint positions $q_{\text{arm}}$; (2) Current gripper action $g_a$, along with sensed gripper joint position $q_{\text{grip}}$; (3) Simulation frequency $H_{\text{sim}}$ (500 in our implementation), action output frequency (control frequency) $H_{\text{ctrl}}$ (5 in our implementation following); (4) Current action timestep $T$ within an episode; (5) A function $S$ that maps a $\mathbb{R}^3$ position vector and a 3x3 $\mathbb{SO}(3)$ rotation matrix to a 4x4 $\mathbb{SE}(3)$ matrix.

1: **if** $T = 0$ **then**                                                   ▷ At the beginning of episode
2:      $q_{\text{lastgoal}} = q_{\text{arm}}$
3: **end if**
4: $(\mathbf{x}, R) = \text{ForwardKinematics}(q_{\text{lastgoal}})$
5: $(\mathbf{x}_{\text{goal}}, R_{\text{goal}}) = S^{-1}(S(\mathbf{x}, I) \cdot S(\mathbf{x}_a, R_a) \cdot$
                        $S(-\mathbf{x}, I) \cdot S(\mathbf{x}, R_{\text{arm}}))$
6: $q_{\text{goal}} = \text{InverseKinematics}(\mathbf{x}_{\text{goal}}, R_{\text{goal}}, q_{\text{arm}})$
7: $q_{\text{goal,grip}} = g_a$
8: $\text{SetArmJointPosTarget}(q_{\text{goal}})$
9: $\text{SetGripperJointPosTarget}(q_{\text{goal, grip}})$
10: $q_{\text{lastgoal}} = q_{\text{goal}}$
11: $T = T + 1$

---

ber of control commands (policy action outputs) per second. We use the open-source library Ruckig[1] for time-optimal joint motion planning with velocity, acceleration, and jerk constraints. Note that the duration of planned trajectories may exceed the interval between two control commands.

**WidowX** We present our WidowX controller implementation in Algorithm 2.

## B.2 Robot and Object Assets

**Robots** For RT-1 Robot, we convert the publically-released MuJoCo `.mjcf` robot description to URDF robot description. We also refine the collision mesh of the robot base link from the original assets to prevent erroneous mesh penetrations. For WidowX, we directly export the URDF robot descriptions from the official Interbotix repository using ROS. To simulate the RT-1 Robot, we find that the Projected Gauss-Seidel solver in PhysX causes mesh penetration behaviors during the process of object grasping. Thus, we enable the Temporal Gauss-Seidel solver in both SAPIEN and Isaac Sim's simulation backends to produce correct grasping behaviors.

The RT-1 Robot uses a customized egocentric camera mounted on the robot head, while the WidowX + Bridge V2 setup uses a Logitech C920 third-view camera. We use known robot camera intrinsics if possible, and when they are unknown, we obtain them from real evaluation video frames using efficient interactive GUI tools such as fSpy.

**Objects** We adopt the following procedure to obtain object assets. Except creating precise models for articulated objects like cabinets, the process is semi-automatic, does not require extensive manual effort, and typically takes less than 20 minutes.

- Obtain raw 3D object models from public repositories (e.g., Objaverse [57]), from 3D scanning of objects purchased from Amazon, from single-view 3D generation (e.g., One-2-3-45++ [61]), or from manual modeling based on precise measurements of real-world counterparts (we only used the last technique for articulated objects like cabinets since this requires the most human effort; we highlight the acceleration of articulated asset curation process through approaches like multi-view [62] or interactive [63] articulated object generation as an avenue for future work).

- Process 3D object models in Blender such that the dimensions of objects are similar to those used in the real world, and that the object meshes do not contain too many vertices (to limit the sizes of object meshes).

- Optionally, use our Visual Matching approach (see below) to improve the texture of 3D object models.

---

[1] https://github.com/pantor/ruckig

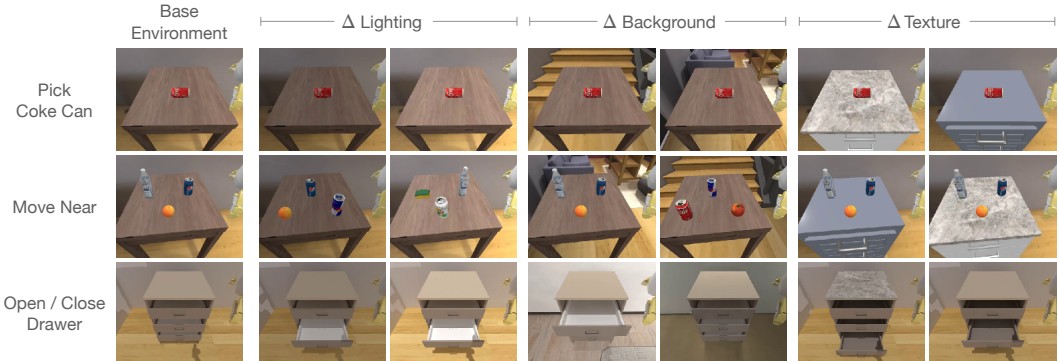

Figure 5: Subset of environment variations under our "Variant Aggregation" evaluation setup, visualized in SAPIEN from RT-1 Robot's egocentric view. The variations cover different lightings, backgrounds and table textures and are modified from ReplicaCAD [16] scenes.

- Export visual mesh and collision mesh of objects. For collision mesh, further perform CoACD [64] to obtain watertight and locally convex collision meshes. Optionally, simplify the resulting collision mesh and perform minor modifications using Blender (e.g., make the bottom of cans or bottles flat).

- Set the object to have a simple uniform density by querying their common material density in GPT-4 or google search, or (for objects with non-uniform densities like empty coke can), querying their mass and dividing by their visual mesh volume.

To perform visual matching of object textures, we adopt the following steps: (1) Crop the target object in a real image using SAM [56]; (2) Perform a *coarse* estimation of object pose by importing it into the simulation and adjusting its position such that its simulation segment mask overlaps with the real one; (3) Employ differential rendering (using Nvdiffrast) to optimize the simulation asset's pose such that it *precisely aligns* with the real object's segmentation mask; (4) "Unproject" the real object's RGB texture values onto the simulation object mesh; (5) Optionally, generate the remaining views of the object through a diffusion model (Zero123++ [65]), and refine the poses of novel views using a rendering loss with the existing object view. Finally, unproject the novel view textures onto the simulation object mesh. This whole process is semi-automatic, and can thus be completed efficiently. We commit to release a convenient command-line python script for this process.

## B.3 SIMPLER-Variant Aggregation

A common approach for addressing visual gaps in sim-to-real policy training is domain randomization. By performing training across a range of randomized parameters, such as textures and lighting, prior works aim to obtain policies that are robust to visual distribution shifts in the real-world [40, 41]. Similarly, in real-to-sim evaluation, we can aggregate evaluation results across a range of visual simulator characteristics to obtain a more faithful signal for the policy's performance. In practice, we implement this SIMPLER-"Variant Aggregation" approach as an alternative to SIMPLER-Visual Matching, described in Section 4.2. Concretely, we create a "base" version of our simulation environment and then creating "variants" of this environment along four axes of visual variation: background, lighting, distractors, and table texture. For each axis, we construct 2 variations of the base setup similar to [60], covering backgrounds from different rooms, lighter and darker lighting, fewer and more distractors, and solid color and complex table textures. We visualize an example of such simulator variations for various table-top tasks in Fig. 5. We average policy performance in simulation across all variants of an environment to obtain our final performance estimate.

| RT-1 Robot Evaluation Setup | Policy | Pick Coke Can | | | | Move Near | Open / Close Drawer | | | Open Top Drawer and Place Apple |
|---|---|---|---|---|---|---|---|---|---|---|
| | | Horizontal Laying | Vertical Laying | Standing | Average | Average | Open | Close | Average | Average |
| Real Eval | RT-1 (Converged) | 0.960 | 0.880 | 0.720 | 0.853 | 0.633 | 0.815 | 0.926 | 0.870 | 0.185 |
| | RT-1 (15%) | 1.000 | 0.960 | 0.800 | 0.920 | 0.583 | 0.704 | 0.889 | 0.796 | 0.185 |
| | RT-1-X | 0.880 | 0.560 | 0.840 | 0.760 | 0.450 | 0.519 | 0.741 | 0.630 | 0.407 |
| | RT-2-X | 0.920 | 0.800 | 1.000 | 0.907 | 0.733 | 0.333 | 0.630 | 0.481 | 0.074 |
| | Octo-Base | 0.440 | 0.200 | 0.240 | 0.293 | 0.350 | 0.148 | 0.519 | 0.333 | 0.000 |
| | OpenVLA-7B | 0.640 | 0.280 | 0.360 | 0.427 | 0.667 | 0.111 | 0.148 | 0.130 | 0.000 |
| | RT-1 (Begin) | 0.200 | 0.000 | 0.200 | 0.133 | 0.017 | 0.000 | 0.000 | 0.000[2] | 0.000 |
| SIMPLER Eval (Variant Aggregation) | RT-1 (Converged) | 0.969 | 0.760 | 0.964 | 0.898 | 0.500 | 0.270 | 0.376 | 0.323 | 0.026 |
| | RT-1 (15%) | 0.920 | 0.704 | 0.813 | 0.813 | 0.446 | 0.212 | 0.323 | 0.267 | 0.021 |
| | RT-1-X | 0.569 | 0.204 | 0.698 | 0.490 | 0.323 | 0.069 | 0.519 | 0.294 | 0.101 |
| | RT-2-X | 0.822 | 0.754 | 0.893 | 0.823 | 0.792 | 0.333 | 0.372 | 0.353 | 0.206 |
| | Octo-Base | 0.005 | 0.000 | 0.013 | 0.006 | 0.031 | 0.000 | 0.021 | 0.011 | 0.000 |
| | OpenVLA-7B | 0.644 | 0.218 | 0.729 | 0.530 | 0.469 | 0.148 | 0.164 | 0.156 | 0.000 |
| | RT-1 (Begin) | 0.022 | 0.013 | 0.031 | 0.022 | 0.040 | 0.005 | 0.132 | 0.069 | 0.000 |
| | **MMRV↓** | 0.149 | 0.194 | 0.240 | 0.168 | 0.105 | 0.350 | 0.328[3] | 0.304 | 0.127 |
| | **Pearson $r$↑** | 0.927 | 0.937 | 0.824 | 0.922 | 0.881 | 0.583 | 0.648 | 0.728 | 0.313 |
| SIMPLER Eval (Visual Matching) | RT-1 (Converged) | 0.960 | 0.900 | 0.710 | 0.857 | 0.442 | 0.601 | 0.861 | 0.730 | 0.065 |
| | RT-1 (15%) | 0.860 | 0.790 | 0.480 | 0.710 | 0.354 | 0.463 | 0.667 | 0.565 | 0.130 |
| | RT-1-X | 0.820 | 0.330 | 0.550 | 0.567 | 0.317 | 0.296 | 0.891 | 0.597 | 0.213 |
| | RT-2-X | 0.740 | 0.740 | 0.880 | 0.787 | 0.779 | 0.157 | 0.343 | 0.250 | 0.037 |
| | Octo-Base | 0.210 | 0.210 | 0.090 | 0.170 | 0.042 | 0.009 | 0.444 | 0.227 | 0.000 |
| | OpenVLA-7B | 0.310 | 0.030 | 0.190 | 0.177 | 0.492 | 0.250 | 0.574 | 0.412 | 0.000 |
| | RT-1 (Begin) | 0.050 | 0.000 | 0.030 | 0.027 | 0.050 | 0.000 | 0.278 | 0.139 | 0.000 |
| | **MMRV↓** | 0.023 | 0.046 | 0.046 | 0.027 | 0.095 | 0.069 | 0.265 | 0.177 | 0.000 |
| | **Pearson $r$↑** | 0.963 | 0.951 | 0.948 | 0.969 | 0.864 | 0.914 | 0.674 | 0.823 | 0.973 |

Table 2: Real-world and SAPIEN evaluation results across different policies on RT-1 Robot tasks. We present success rates for the "Variant Aggregation" and "Visual Matching" approaches in Sec. 4.2. We calculate the Mean Maximum Rank Violation ("MMRV", lower is better) and the Pearson correlation coefficient ("Pearson $r$", higher is better) to assess the alignment between simulation and real-world relative performances across different policies.

| WidowX+Bridge Evaluation Setup | Policy | Put Spoon on Towel | | Put Carrot on Plate | | Stack Green Block on Yellow Block | | Put Eggplant in Yellow Basket | |
|---|---|---|---|---|---|---|---|---|---|
| | | Grasp Spoon | Success | Grasp Carrot | Success | Grasp Green Block | Success | Grasp Eggplant | Success |
| Real Eval | RT-1-X | 0.042 | 0.000 | 0.167 | 0.000 | 0.000 | 0.000 | 0.033 | 0.000 |
| | Octo-Base | 0.500 | 0.333 | 0.500 | 0.250 | 0.292 | 0.000 | 0.400 | 0.233 |
| | Octo-Small | 0.542 | 0.417 | 0.208 | 0.083 | 0.583 | 0.125 | 0.700 | 0.433 |
| SIMPLER Eval (Visual Matching) | RT-1-X | 0.167 | 0.000 | 0.208 | 0.042 | 0.083 | 0.000 | 0.000 | 0.000 |
| | Octo-Base | 0.347 | 0.125 | 0.528 | 0.083 | 0.319 | 0.000 | 0.667 | 0.431 |
| | Octo-Small | 0.778 | 0.472 | 0.278 | 0.097 | 0.403 | 0.042 | 0.875 | 0.569 |
| | **MMRV↓** | 0.000 | 0.000 | 0.000 | 0.111 | 0.000 | 0.000 | 0.000 | 0.000 |
| | **Pearson $r$↑** | 0.778 | 0.827 | 0.995 | 0.575 | 0.964 | 1.000 | 0.995 | 0.990 |

Table 3: Real-world and SAPIEN simulation evaluation results for the WidowX + Bridge setup. We report both the final success rate ("Success") along with partial success (e.g., "Grasp Spoon").

# C  Full Results for Real-and-Sim Relative Policy Performance Correlation Experiments

In Table 2 and Table 3, we present full evaluation results for our experiments in Sec. 5.2, which demonstrate that SIMPLER environments show strong performance relationship correlations with real-world evaluations.

| Policy | Distribution Shift | Pick Coke Can | | | Move Near | | | Avg. | | | Real TableTop [60] |
|---|---|---|---|---|---|---|---|---|---|---|---|
| | | \|Δ Success\| | MMRV↓ | $r$ ↑ | \|Δ Success\| | MMRV↓ | $r$ ↑ | \|Δ Success\| | MMRV↓ | $r$ ↑ | \|Δ Success\| |
| RT-1 w/o Aug | Background | 0.013 | | | 0.083 | | | 0.048 | | | 0.028 |
| | Lighting | 0.040 | | | 0.075 | | | 0.057 | | | 0.083 |
| | Distractors | 0.027 | 0.000 | 0.779 | 0.133 | 0.055 | 0.939 | 0.080 | 0.000 | 0.831 | 0.111 |
| | Table Texture | 0.113 | | | 0.175 | | | 0.144 | | | 0.389 |
| | Camera Pose | 0.753 | | | 0.192 | | | 0.473 | | | 0.458 |
| RT-1 +Aug | Background | 0.153 | | | 0.092 | | | 0.123 | | | 0.167 |
| | Lighting | 0.033 | | | 0.117 | | | 0.075 | | | 0.042 |
| | Distractors | 0.033 | 0.041 | 0.984 | 0.084 | 0.125 | 0.721 | 0.059 | 0.041 | 0.970 | 0.083 |
| | Table Texture | 0.220 | | | 0.159 | | | 0.189 | | | 0.167 |
| | Camera Pose | 0.613 | | | 0.175 | | | 0.394 | | | 0.375 |

Table 4: Impact of various distribution shifts on the tabletop manipulation performance of RT-1 policies trained with and without image augmentation. SIMPLER evaluations accurately track the policies' robustness to distribution shifts, exhibiting low Mean Maximum Rank Violation ("MMRV") and high Pearson correlation coefficient ("$r$") with the real world evaluations [60].

| Policy | Robustness Factor | Pick Coke Can | Move Near |
|---|---|---|---|
| RT-1 w/o Aug | Base Setup | 0.920 | 0.467 |
| | Background | 0.933/0.907 | 0.533/0.567 |
| | Lighting | 0.960/0.960 | 0.483/0.600 |
| | Distractors | 0.880/0.901 | 0.600[a] |
| | Table Texture | 0.867/0.747 | 0.550/0.200 |
| | Camera Pose | 0.053/0.280 | 0.117/0.433 |
| RT-1 +Aug | Base Setup | 0.800 | 0.383 |
| | Background | 0.747/0.547 | 0.483/0.467 |
| | Lighting | 0.760/0.773 | 0.517/0.483 |
| | Distractors | 0.813/0.747 | 0.467 |
| | Table Texture | 0.667/0.493 | 0.450/0.133 |
| | Camera Pose | 0.267/0.107 | 0.200/0.217 |

[a]The base setup environment already contains distractors, so we construct environment variants without distractors.

Table 5: Success rates of different out-of-distribution generalization factors on the tabletop manipulation performance of RT-1 policies in the SAPIEN simulator. "a/b" denote results on different environment variants (lighting: darker / brighter; table texture: solid color / contrastively patterned; camera pose: oriented lower / higher).

# D Full Results for Real-and-Sim Policy Behavior Correlation Experiments under Environment Distribution Shifts

In Table 4, Table 5, and Table 6, we present full evaluation results for our experiments in Sec. 5.3, which demonstrate that SIMPLER environments show strong policy behavior correlations with real-world evaluations under different environment distribution shifts.

# E Full Results for Main Paper Ablation Experiments

We present detailed results for our main paper's ablations in Table 7, Table 8, and Table 9.

[2]After running 2 real evaluation trials, robot operators decided that since this policy would potentially damage the robot on the Drawer tasks, the real evaluation was terminated.

[3]As real evaluation was terminated due to risk of damaging the robot, we expect the MMRV to be less than this number if real evaluation were to continue.

| Policy | Sim Success Range | Real Success | |
|---|---|---|---|
| | | Orig Arm Texture | OOD Arm Texture |
| RT-1-X | [0.507, 0.653] | 0.760 | 0.520 |
| Octo-Base | [0.000, 0.293] | 0.293 | 0.000 |

Table 6: Impact of arm textures on the success rates of "Pick Coke Can" task in the SAPIEN simulator (Visual Matching evaluation setup) and in the real-world. The ranges of simulation success rates across multiple (tuned and untuned) robot arm colors can predict policy sensitivity to real-world OOD arm textures.

| Components | | | Open Drawer | | | | Close Drawer | | | |
|---|---|---|---|---|---|---|---|---|---|---|
| Background | Drawer | Robot | RT-1 (Converged) | RT-1 (15%) | RT-1-X | MMRV↓ | RT-1 (Converged) | RT-1 (15%) | RT-1-X | MMRV↓ |
| Real | Real | Real | 0.815 | 0.704 | 0.519 | N/A | 0.926 | 0.889 | 0.741 | N/A |
| GreenScreen | Curated | Curated | **0.703** | **0.556** | **0.333** | **0.000** | **0.889** | 0.667 | 0.851 | **0.099** |
| GreenScreen | Curated | Original | 0.444 | 0.444 | 0.259 | 0.111 | 0.741 | 0.630 | 0.926 | 0.173 |
| GreenScreen | Original | Original | 0.593 | 0.519 | 0.148 | **0.000** | 0.852 | **0.778** | 0.963 | 0.173 |
| ReplicaCAD | Curated | Curated | 0.407 | 0.259 | 0.111 | **0.000** | 0.667 | 0.481 | 0.778 | 0.173 |
| ReplicaCAD | Curated | Original | 0.630 | 0.407 | 0.074 | **0.000** | 0.630 | 0.593 | 0.667 | 0.173 |
| ReplicaCAD | Original | Curated | 0.556 | 0.296 | 0.074 | **0.000** | 0.667 | 0.704 | 0.815 | 0.173 |
| ReplicaCAD | Original | Original | 0.556 | 0.333 | 0.074 | **0.000** | 0.704 | 0.556 | **0.741** | 0.173 |

Table 7: Impact of real-to-sim visual gaps on real-and-sim performance correlations. We report the success rates of 3 different policies on 2 tasks: *Open Drawer* and *Close Drawer*. The settings with the smallest MMRV and the smallest absolute performance gap with the real performance are highlighted. Using a combination of "green-screened" background and curated foreground object and robot assets provides the best correlation.

| | Gripper Friction Coefficient | | | |
|---|---|---|---|---|
| Coke Can Mass | 0.25 | 0.50 | 1.0 | 2.0 |
| 10 g | 0.957 | 0.967 | 0.971 | 0.978 |
| 20 g | 0.969 | 0.975 | 0.978 | 0.977 |
| 40 g | 0.963 | 0.976 | 0.976 | 0.976 |
| 80 g | 0.962 | 0.962 | 0.975 | 0.990 |

(a) Pearson $r$ between real and SIMPLER evaluations on the Pick Coke Can task under different settings of can mass and gripper friction coefficient. The MMRV is 0.031 in all cases. The use of empty coke cans follows the setup from the RT-1 Robot demonstration dataset and the RT-1 paper [1].

| Cabinet Joint Friction | 0.0125 | 0.025 | 0.05 | 0.10 | 0.15 | 0.20 |
|---|---|---|---|---|---|---|
| MMRV↓ | 0.055 | 0.055 | 0.055 | 0.055 | 0.105 | 0.055 |
| Pearson $r$↑ | 0.930 | 0.941 | 0.915 | 0.923 | 0.903 | 0.928 |

(b) MMRV and Pearson $r$ between real and SIMPLER evaluations on the Open/Close Drawer tasks under different settings of cabinet joint friction.

Table 8: SIMPLER is robust to imprecisely estimated physical simulation parameters such as object mass and friction coefficients, as indicated by the low MMRV and high Pearson $r$ in both ablation studies. We use the 6 policies from our RT-1 Robot experiments in these ablations.

| RT-1 Robot Evaluation Setup | Policy | Pick Coke Can | | | | Move Near |
|---|---|---|---|---|---|---|
| | | Horizontal Laying | Vertical Laying | Standing | Avg. Success | Avg. Success |
| Real Eval | RT-1 (Converged) | 0.960 | 0.880 | 0.720 | 0.853 | 0.633 |
| | RT-1 (15%) | 1.000 | 0.960 | 0.800 | 0.920 | 0.583 |
| | RT-1-X | 0.880 | 0.560 | 0.840 | 0.760 | 0.450 |
| | Octo-Base | 0.440 | 0.200 | 0.240 | 0.293 | 0.350 |
| | RT-1 (Begin) | 0.200 | 0.000 | 0.200 | 0.133 | 0.017 |
| SIMPLER Eval (Isaac, Variant Aggre.) | RT-1 (Converged) | 0.418 | 0.377 | 0.436 | 0.410 | 0.150 |
| | RT-1 (15%) | 0.428 | 0.306 | 0.590 | 0.441 | 0.100 |
| | RT-1-X | 0.340 | 0.182 | 0.618 | 0.380 | 0.125 |
| | Octo-Base | 0.015 | 0.020 | 0.010 | 0.015 | 0.020 |
| | RT-1 (Begin) | 0.036 | 0.040 | 0.054 | 0.044 | 0.000 |
| | **MMRV↓** | 0.096 | 0.112 | 0.016 | 0.064 | 0.053 |
| | **Pearson $r$↑** | 0.961 | 0.949 | 0.989 | 0.973 | 0.865 |

Table 9: Real-world and Isaac Sim evaluation results for the RT-1 Robot setup. The findings on Isaac Sim are consistent with the findings on the SAPIEN simulator.

| Task | Validation Action MSE | Sim Eval (Visual Matching) |
|------|------------------------|-----------------------------|
| Pick Coke Can | 0.412 / 0.464 | 0.031 / 0.976 |
| Move Near | 0.408 / 0.230 | 0.111 / 0.855 |
| Open / Close Drawer | 0.346 / 0.264 | 0.055 / 0.915 |
| Open Drawer & Place Apple | 0.265 / 0.198 | 0.000 / 0.969 |
| Put Spoon on Towel | 0.389 / -0.951 | 0.000 / 0.827 |
| Put Carrot on Plate | 0.194 / -0.342 | 0.111 / 0.575 |
| Stack Block | 0.125 / -0.857 | 0.000 / 1.000 |
| Put Eggplant in Basket | 0.366 / -1.000 | 0.000 / 0.990 |

Table 10: MMRV / Pearson correlation comparison between our Visual Matching simulation evaluation approach and the simulation-free approach that assesses the MSE between predicted and ground-truth actions on validation trajectories. For the latter approach, we calculate the MMRV / Pearson correlation between the negative MSE and the real policy performance. Our approach yields significantly better real-and-sim policy performance correlations.

| Policy | Avg. Real Success | Avg. Sim Success (Visual Matching) |
|--------|-------------------|-------------------------------------|
| RT-1 (Converged) | 0.853 | 0.857 |
| RT-1 (15%) | 0.920 | 0.710 |
| RT-1 (Single Task Policy) | 0.680 | 0.403 |
| RT-1-X | 0.760 | 0.567 |
| RT-2-X | 0.907 | 0.787 |
| Octo-Base | 0.293 | 0.170 |
| RT-1 (Begin) | 0.133 | 0.027 |
| **MMRV↓** | | 0.027 |
| **Pearson** $r$↑ | | 0.959 |

Table 11: Real-world and simulated evaluation results on the Pick Coke Can task, after adding an RT-1 policy trained solely on the Pick Coke Can demonstrations. Our simulated evaluation remains effective, exhibiting low MMRV and high Pearson correlation coefficient with real evaluations.

# F   More Experiment Results

## F.1   More Ablations

**Simulated vs. simulation-free evaluation approaches**: To evaluate and select real-world robot manipulation policies, a widely-adopted approach involves calculating the MSE loss between predicted and ground-truth actions on a set of held-out validation demonstration trajectories. We are thus interested in the following question: *Does simulated evaluation produce significantly better real-to-sim relative performance correlation than simulation-free approaches?* We conduct an experiment where we calculate the action-prediction MSE loss on the RT-1 Robot dataset and the Bridge dataset. For the Bridge dataset, we randomly select 25 trajectories from the validation demonstration split. For the RT-1 Robot dataset, as a validation split is not publicly available, we randomly select 25 trajectories from the training demonstrations.

We report the results in Table 10. We find that SIMPLER evaluation produces significantly better correlations between real-and-sim performances across different policies, as highlighted by a substantially-lower MMRV and a substantially-higher Pearson correlation coefficient. Furthermore, as demonstrated in Sec. 5.3 of the main paper, SIMPLER evaluation reveals finegrained policy behavior modes, such as robustness to visual distribution shifts, offering insights beyond policy performance comparisons, unlike simulation-free evaluations.

**Is simulated evaluation still effective on single-task policies?** Previously in the main paper, we focused our simulated evaluation on policies trained on multi-task datasets, such as the RT-1 Robot RT-1 dataset and the Open-X-Embodiment dataset, which contain ≥80k demonstrations. In this section, we further ask the question: *Is SIMPLER evaluation still effective on policies trained on smaller-scale data, which are potentially more sensitive to real-to-sim visual and control gaps?* To this end, we conduct an experiment where we train RT-1 only with the "pick coke can" demonstrations and evaluate its real and simulation performance. We also compare the MMRV and the

| RT-1 Robot Evaluation Setup | Metric | Pick Coke Can | | | | Move Near | Open / Close Drawer | | | Open Top Drawer and Place Apple |
|---|---|---|---|---|---|---|---|---|---|---|
| | | Horizontal Laying | Vertical Laying | Standing | Avg. Success | Avg. Success | Open | Close | Avg. Success | Avg. Success |
| SIMPLER (VisMatch) | Kruskal-#Policy p<0.05 | 0 | 0 | 2 | 3 | 3 | 1 | 2 | 2 | 0 |

(a)

| WidowX+Bridge Evaluation Setup | Metric | Put Spoon on Towel | | Put Carrot on Plate | | Stack Green Block on Yellow Block | | Put Eggplant in Yellow Basket | |
|---|---|---|---|---|---|---|---|---|---|
| | | Grasp Spoon | Success | Grasp Carrot | Success | Grasp Green Block | Success | Grasp Eggplant | Success |
| SIMPLER (VisMatch) | Kruskal-#Policy p<0.05 | 0 | 0 | 0 | 0 | 0 | 0 | 1 | 0 |

(b)

Table 12: For our Visual Matching evaluation approach, we conduct Kruskal-Wallis test to assess whether simulation and real-world policy evaluations exhibit significant distribution shift, even though we do not expect to obtain an exact reproduction of real-world performance.

Pearson correlation before and after incorporating this single-task policy into the RT-1 Robot experiments. Results are shown in Table 11. We find that our simulated evaluation effectively reflects the performance rankings of the newly-added single-task policy, with the MMRV remaining low and the Pearson Coefficient remaining high. This demonstrates SIMPLER evaluation's versatility across policies trained on diverse data scales.

### F.2 Other Metrics: Kruskal Wallis

In our previous analysis, we primarily focused on metrics that measure real-to-sim **relative performance** alignment between policies. As we match real-to-sim visual input appearance in our Visual Matching evaluation approach, it also becomes meaningful to measure the simulation distribution shift of **absolute performance** from real-world evaluations, even though we do not expect the real-to-sim absolute performances to exactly match. Let the real-world evaluation results of $N$ policies be $\mathbf{r} = \{\mathbf{r}_1, \mathbf{r}_2, \ldots, \mathbf{r}_N\}$, where $\mathbf{r}_i = (r_{ij})_{j=1}^{N_{\text{trial}}}$ is the indicator of each trial's success in the real-world. Let the corresponding simulation evaluation results be $\mathbf{s} = \{\mathbf{s}_1, \mathbf{s}_2, \ldots, \mathbf{s}_N\}$, where $\mathbf{s}_i = (s_{ij})_{j=1}^{N_{\text{trial}}}$. We perform *Kruskal-Wallis test* for each individual policy (i.e., between each $\mathbf{r}_i$ and $\mathbf{s}_i$) to measure whether simulation evaluations exhibit significant distribution shift from real evaluations. We then report the number of policies with significant distribution shift (which we denote as "Kruskal-#Policy p<0.05").

We present the Kruskal-Wallis results in Tab. 12. We find that with the Visual Matching evaluation approach, the simulation trial success distribution is not significantly different from the real results ($p \geq 0.05$) across many tasks and policies, demonstrating the effectiveness of our simulation evaluation tool. We also note that our MMRV and the Kruskal metrics complement each other's limitations, with the former providing a real-to-sim relative performance alignment perspective, and the latter providing an absolute performance alignment perspective.

## G  Limitations of Our Current Work

Our current work has several limitations: **(1)** We focus our evaluations on rigid-object manipulation tasks, as they are most straightforward to simulate. Extending simulated evaluation to soft-object tasks is an exciting avenue for future work. **(2)** For our Visual Matching evaluation approach, the background "green-screening" technique limits evaluations to fixed cameras and does not accurately capture object shadows and other visual details (our Variant Aggregation evaluation does support variable camera poses). **(3)** Our current pipeline for generating simulated evaluation environments involves some manual effort in curating assets and assembling scenes. Enabling a fully automatic and more scalable pipeline for creating thousands of realistic simulated environments is an ambitious goal for future work. In particular, a promising future direction is to automatically reconstruct real-world scenes and backgrounds with high-quality geometries, materials, and lighting, while enabling

fast simulation of these scenes. In the beginning of the project, we did preliminary investigations using state-of-the-art approaches in Gaussian Splatting [66, 67], but found the scene quality to be insufficient for real-to-sim manipulation evaluation (e.g., uneven table surface geometries, floaters while rendering from novel camera views). Additionally, importing Gaussian Splatting-based scenes into simulation environments would entail non-trivial changes to the existing simulators' rendering pipeline, which we leave for future work. **(4)** Our current simulated environments are limited to RGB image and proprioceptive policy inputs. We believe that future simulated evaluation tools could incorporate other modalities such as LIDAR [6], acoustic [68], and tactile sensors [69] as research on their effective and accurate simulation continues to advance. **(5)** We adopt a simple system identification approach that we find to be adequate for scenarios like rigid object grasping and articulated furniture manipulation. It is unlikely that system identification using open-loop contact-free trajectories will be sufficient to accurately model robot control parameters for situations such as high-speed robot-object collisions and dexterous manipulation.

