# OpenReview forum: "Evaluating Real-World Robot Manipulation Policies in Simulation"
_robot-learning.org/CoRL/2024/Conference — CoRL 2024_

### Official Review · Reviewer_mzQD · 2024-07-19
**Paper Review**

**Originality:** 4
**Technical Quality:** 4
**Clarity Of Presentation:** 4
**Potential Impact:** 3
**Recommendation:** 3
**Confidence:** 4

**Review:**

### Strengths

1. The paper tackles an important and long-standing challenge of reproducible evaluation of robotic policies. In contrast to prior works, the paper does not assume a fixed set of evaluation objects or training in simulation, and is applicable to policies pre-trained on real world data.

2. I found the paper to be well motivated and enjoyable to read. The setup is well motivated and the approach is clearly presented and easy to follow.

3. The experimental section evaluates key aspects of the proposed evaluation environments - correlation with real world performance as well as testing finer grained details of policy behavior such as distribution shifts.


### Weaknesses

1. In the related work section, the authors do not elaborate sufficiently on existing methods for real-to-sim.

2. It is unclear how the current approach will scale to real world policies with more complicated inputs like tactile sensors which can be hard to simulate at high level of accuracy. The authors should comment on how this could potentially be achieved or mention it in the limitations.

**Quality Of The Limitations Section:**

2

**Questions For Rebuttal:**

Q1. Sec 3: it is unclear how the ranking metric behaves if the simulated environment consistently overestimates the real-world performance of policies. In this case, even though we have a strong correlation, simulated evaluations could mislead us to believe that a bad real-world policy is good. How does the current ranking metric overcome this issue?

Q2. Sec 4.1 :  The offline system identification approach does not clearly address the coupling between end-effector motion and object properties when in contact. Does the offline dataset used for system identification also span different objects being manipulated or other potential contact surfaces in the environment?

**Robotics Focus:**

4

**Summary Of Paper:**

This paper explores the challenging problem of simulation-based evaluation of robot manipulation policies trained from real world data. The key insight is to develop realistic simulated environments that are realistic enough by overcoming control and visual disparities, so they can serve as a scalable and repeatable tool for evaluating real-world manipulation policies. The paper proposes different simulated environments and a ranking metric for policy evaluation showing strong correlation in policy performance between real-world and sim for open-source RT1-X and Octo manipulation policies.

**Summary Of Recommendation:**

I am recommending weak accept based on the points mentioned in weaknesses and questions above. I am willing to update my recommendation if authors sufficiently address them.

---

### Official Review · Reviewer_VgL2 · 2024-07-21
**Well written and valuable contributions**

**Originality:** 3
**Technical Quality:** 4
**Clarity Of Presentation:** 5
**Potential Impact:** 4
**Recommendation:** 4
**Confidence:** 4

**Review:**

Strengths
1. The paper’s contributions are valuable: the proposed framework takes a step towards reproducible benchmarking of general purpose manipulation policies in simulation, demonstrating strong correlation with real robot performance.
2. The different choices are well analysed and it is interesting to see the simulation environments discovering novel distribution shifts: sensitivity to robot arm textures and patterned tables.
3. The paper is well written and the procedure for generating simulated environments is well detailed in the paper and the appendix.

Weaknesses
1. The real2sim conversion of assets and textures seem to require multiple steps of human effort. Raw estimates of manual hours needed for creating a single evaluation environment would be useful to include.
2. The diversity and scale of evaluation environments (backgrounds, textures, static camera angles) is limited to the evaluation environments present in the real world trajectories (due to paired creation). While some axes of further variation are explored (eg. arm trajectories), further scaling of unpaired simulated evaluation environments is unexplored.
3. Performance in simulation seems to be slightly uncorrelated with real robot performance in select scenarios (Close Drawer Table 2, Grasp Spoon Table 3, Augs not helping Move Near in Table 4).

**Quality Of The Limitations Section:**

3

**Questions For Rebuttal:**

Elaborated under weaknesses. Critically would like to know an estimate of manual hours spent preparing the environments, and cause of poor correlation in select cases.

**Robotics Focus:**

4

**Summary Of Paper:**

The authors propose a simulation framework, called SIMPLER, for evaluating real-robot manipulation policies at scale. The authors propose to close the sim-real control gap via system identification using a small sample of real trajectories and the sim-real appearance gap by matching real backgrounds and textures. The simulation performance is shown to strongly correlate with real robot performance, and is shown to be indicative of the relative sensitivity of real robot policies to different distribution shifts.

**Summary Of Recommendation:**

Proposing acceptance as the developed framework holds value for reproducible benchmarking of real robot manipulation policies.

---

### Official Review · Reviewer_SfVk · 2024-07-25
**Review of submission 9**

**Originality:** 2
**Technical Quality:** 3
**Clarity Of Presentation:** 3
**Potential Impact:** 2
**Recommendation:** 3
**Confidence:** 4

**Review:**

Summary

This paper presents a framework called SIMPLER for validating policies in simulation that have been trained with real robot evaluation. The methodology in the paper proceeds by to estimating empirically the correlation between the simulation evolution and real-world evaluation of the policies. The key contributions are an algorithmic framework for assessing the performance of such policies in simulation that highly correlates with the assessment in evaluation on the real robotic platform. Mainly the authors propose two features that are essential for robotic manipulation tasks. First, importing textures from the real dataset to the simulator, and second, performing system identification for having a robot behavior that mimics the real one.

Review

= Contribution

The paper’s contribution is focused on robot manipulation tasks. It also assumes that the policies are optimized for a certain distribution. In this context the simulation scenarios, are designed by hand and replicate some of the available scenarios in the dataset. In contrast, the recent trend in autonomous driving and other robotic applications is to automate this step using data-driven simulation. In a nutshell, this means that the dataset is automatically used for generating novel scenario that are generalized using hybrid modeling. In my humble opinion, the approach of the paper is more incremental and probably performs better in some cases but requires a lot of manual design.

= Research Idea

== Quality

The paper is very well executed. The experiments are and the focus are timely. The contribution is clear but the situation with respect to state-of-the-art in policy validation is not clear.

It would be nice to have a broader picture in terms of the limitations and tradeoffs of the approach. What is generally the challenge in simulation? Is it the model accuracy or the compute performance? Are there other challenges linked to contact simulation, which are specific to manipulation? Linked to model-based reinforcement learning? How difficult is it learn a closed-loop version of an open-loop dataset? What are the perspectives in the domain?

== clarity

The paper is quite clear and well written. The experiments clearly show that the aim of the paper are met and that the focus of SIMPLER (Visual matching and System Identification) are well chosen. In the limitation part of the paper the authors show that they are conscious of the need for more automation.

== originality

The contribution is interesting in the scene that it clearly addresses the problem of validating real world policies in simulation and succeeds to that end. As mentioned above this type of approach might be very well suited for this problem. The limitation to camera poses may however be a showstopper for real world applications.

== significance

The significance of the work I would rate as low. It is a timely contribution because the community needs to work on full statistical assessment of these foundation models for robots. However, it is important to have the ability to assess out-of-distribution scenarios, and to modulate how far from distribution the validation takes place. The first thing is to be able to automatically generate simulation pipeline. Hence the research effort should be placed in designing fully automated frameworks.

= Strength and Weakness

Strengths
-	Addressing a timely problem
-	Very well written paper & clarity of argumentation
-	Interesting idea and theoretical considerations
-	Good baseline comparison

Weakness
-	Not sufficiently connection to state-of-the-art methods (i.e., NeRFs etc)
-	Not enough focus on tackling the difficult scientific issue linked to the problem
-	Limited to in distribution testing with very moderate out-of-distribution validation

**Quality Of The Limitations Section:**

3

**Questions For Rebuttal:**

There are no questions from my side.

**Robotics Focus:**

3

**Summary Of Paper:**

This paper presents a framework called SIMPLER for validating manipulation policies in simulation that have been trained with real robot evaluation.

**Summary Of Recommendation:**

I would be ok to accept this paper if others fight for it. But my recommendation is rather borderline (which is not possible here).

---

### Author Rebuttal · Authors · 2024-08-07

Thanks all reviewers and the AC for the constructive feedback! We have revised our manuscript accordingly in the attached file below.

---

### Decision · Program_Chairs · 2024-09-04

**Decision:**

Accept

**Comment:**

This paper appears to have been relatively well-received by the reviewers, with a number of notable strengths.

-well written and well-structured
-clear in its motivation
-timely
-algorithm and approach is well-defined and well-described

But with some weaknesses:

-apparent use of extensive manual labour in capturing textures etc. (reviewer VgL2)
-lack of thorough discussion on weaknesses and limitations
-uses only camera poses
-unclear sim2real transferability according to some of the reported results, which requires some investigation and explanation
-does not consider some SOTA techniques, e.g., NERFs.

The authors should clarify these issues
I encourage the authors to address the comments of the reviewers to increase chances of acceptance.


Updated:

The authors have been responsive to the reviewer's comments, run extra experiments, and tidied up the text.